# Angiotensin Receptor-Neprilysin Inhibitor (ARNI) and Cardiac Arrhythmias

**DOI:** 10.3390/ijms22168994

**Published:** 2021-08-20

**Authors:** Henry Sutanto, Dobromir Dobrev, Jordi Heijman

**Affiliations:** 1Department of Cardiology, CARIM School for Cardiovascular Diseases, Maastricht University, 6229 ER Maastricht, The Netherlands; henry.sutanto@maastrichtuniversity.nl; 2Department of Physiology and Pharmacology, State University of New York (SUNY) Downstate Health Sciences University, Brooklyn, NY 11203, USA; 3West German Heart and Vascular Center, Institute of Pharmacology, University Duisburg-Essen, 45147 Essen, Germany; dobromir.dobrev@uk-essen.de; 4Montréal Heart Institute, University de Montréal, Montréal, QC H1T 1C8, Canada; 5Department of Molecular Physiology & Biophysics, Baylor College of Medicine, Houston, TX 77030, USA

**Keywords:** neprilysin, renin-angiotensin-aldosterone, arrhythmia, cardiovascular, pharmacology, heart rhythm, electrophysiology, heart failure, natriuretic peptide

## Abstract

The renin-angiotensin-aldosterone system (RAAS) plays a major role in cardiovascular health and disease. Short-term RAAS activation controls water and salt retention and causes vasoconstriction, which are beneficial for maintaining cardiac output in low blood pressure and early stage heart failure. However, prolonged RAAS activation is detrimental, leading to structural remodeling and cardiac dysfunction. Natriuretic peptides (NPs) are activated to counterbalance the effect of RAAS and sympathetic nervous system by facilitating water and salt excretion and causing vasodilation. Neprilysin is a major NP-degrading enzyme that degrades multiple vaso-modulatory substances. Although the inhibition of neprilysin alone is not sufficient to counterbalance RAAS activation in cardiovascular diseases (e.g., hypertension and heart failure), a combination of angiotensin receptor blocker and neprilysin inhibitor (ARNI) was highly effective in several clinical trials and may modulate the risk of atrial and ventricular arrhythmias. This review summarizes the possible link between ARNI and cardiac arrhythmias and discusses potential underlying mechanisms, providing novel insights about the therapeutic role and safety profile of ARNI in the cardiovascular system.

## 1. General Introduction

Pharmacological agents increasing the level of circulating natriuretic peptides (NPs) have been proposed for the management of cardiovascular diseases. Some, including the recently-developed angiotensin receptor-neprilysin inhibitor (ARNI), have shown positive results on patients’ outcome, particularly in those with hypertension and heart failure (HF). However, emerging evidence suggests that ARNI may also modulate the risk of cardiac arrhythmias, with both pro- and antiarrhythmic effects being reported. Here, we review the putative roles of ARNI in cardiovascular disease management, focusing on their potential effects in cardiac arrhythmogenesis, and discuss the potential molecular mechanisms of arrhythmia modulation by ARNI. 

## 2. The Role of Renin-Angiotensin-Aldosterone System (RAAS) and NP in Cardiovascular Pathophysiology

The RAAS plays a key role in the physiological regulation of the renal and cardiovascular systems [1]. In response to several intrinsic stimuli, including a decrease in renal blood pressure, sympathetic nervous system (SNS) stimulation, and reduced sodium delivery in the renal distal convoluted tubule, renal juxtaglomerular cells convert inactive prorenin into renin, which is released into the circulation. Renin then interacts with liver-originated angiotensinogen, enabling the cleavage of angiotensinogen into angiotensin I, which is converted into angiotensin II by the angiotensin-converting enzyme (ACE) [1]. In the adrenal cortex, angiotensin II stimulates the release of aldosterone, which modulates sodium reabsorption and potassium excretion in the kidney [1]. Collectively, stimulation of angiotensin II receptors in various organs is essential to maintain water and electrolytes homeostasis, and to regulate vascular tone during hypotension (Figure 1). RAAS activation plays a major role in several cardiovascular diseases. For example, in HF, the diminished cardiac output activates the RAAS to maintain blood pressure. RAAS activation results in vasoconstriction, water and sodium retention, which are beneficial adaptive mechanisms in early-stage HF. However, prolonged RAAS activation increases pre- and afterload, worsening the left-ventricular (LV) function and promoting cardiac remodeling [2]. 

NPs similarly play an important role in cardiovascular homeostasis [2,3,4]. At least three major subtypes of NPs exist: atrial, brain and C-type NPs (i.e., ANP, BNP and CNP), which are primarily synthesized in, respectively, the atria, ventricles, and vascular endothelium (e.g., in central nervous system and HF myocardium) [2,5]. Initially, these NPs are synthesized and secreted as inactive precursors (in a pre-pro form). During prolonged activation of the RAAS and SNS in HF, the pre-pro-NP is converted into pro-NP, which is subsequently cleaved into active NP, and inactive N-terminal (NT) pro-NP fragments [3]. The active fragments bind to NP receptors (NPRs) to exert their broad array of functions, including natriuresis, diuresis and vasodilation, lowering blood pressure and elevating vascular permeability (Figure 2). Active NPs also inhibit smooth muscle-cell proliferation and the release of renin and aldosterone, thereby reducing the HF-associated structural remodeling [2]. NPs are inactivated by body fluid-mediated excretion (e.g., urine and bile), NPR-mediated internalization (via NP clearance receptors NPR-C and NPR-C3), and proteolysis. Neprilysin is the primary membrane-bound proteolytic enzyme responsible for NP degradation. It is predominantly expressed in the kidney and also digests other vessel-modulating substances (e.g., substance-P, bradykinin, endothelin-1 and angiotensin II), as well as converting angiotensin I into vasodilation-promoting angiotensin 1–7 [2]. Under physiological conditions, the concentration of NPs is inversely correlated with RAAS activation, while in HF, this correlation is positive, with NP levels proportional to the cardiac injury, presumably acting as a compensatory, self-protective mechanism [2]. 

Several pharmacological interventions have been explored to preserve or increase the level of NPs to counterbalance excessive RAAS and SNS activation in cardiovascular diseases. These include the administration of exogenous NPs (e.g., nesiritide and carperitide) and the inhibition of neprilysin-mediated NP degradation (e.g., with racecadotril, candoxatrilat, ecadotril, candoxatril or sacubitril) [3]. Although exogeneous NPs successfully increased circulating NPs and natriuresis, they produced limited improvement of clinical outcomes. In patients with decompensated HF, nesiritide, a recombinant BNP, did not significantly improve mortality, yet it increased the risk for hypotension and bradycardia [6]. Likewise, in patients with hypertension, neprilysin inhibitors only reduced blood pressure transiently, possibly due to reduced neprilysin-mediated endothelin-1 and angiotensin II degradation [2,3]. Meanwhile, increased renal hypertrophy and glomerular lesions were documented in hypertensive rats following the administration of sacubitril, indicating the progression of renal disease with neprilysin inhibition [7]. Therefore, combinations of neprilysin inhibitors with RAAS blockers (i.e., ACE inhibitor [ACEi] or angiotensin receptors blocker [ARB]) were proposed. However, the combination of ACEi and neprilysin inhibitors augmented the ACEi-induced accumulation of bradykinin by reducing neprilysin-mediated bradykinin degradation [3]. Bradykinin promotes vasodilation, inhibits thrombus formation and prevents ischemia, fibrosis, cell proliferation, and hypertrophy, highlighting its cardioprotective effects. However, bradykinin accumulation also mediates local histamine release, bronchospasm, inflammation, hyperalgesia and vascular hyperpermeability, promoting dry cough and angioedema [8]. The combination of ACEi and neprilysin inhibitors indeed effectively lowered angiotensin II level and blood pressure in hypertensive patients but facilitated a higher incidence of bradykinin accumulation-promoted angioedema [9], making the combination of ACEi/neprilysin inhibitor unfavorable. Meanwhile, ARNI (i.e., an ARB combined with a neprilysin inhibitor) has shown multiple positive effects in numerous cardiovascular diseases. 

## 3. The Promising Roles of ARNI in Cardiovascular Disease Management

### 3.1. Systemic Hypertension

ARNIs are considered superior to ACEi/neprilysin inhibitors because they induce less bradykinin accumulation and angioedema. Indeed, the combination of valsartan/candoxatril lowered blood pressure in spontaneous hypertensive rats (SHRs) with less tracheal plasma extravasation, a marker of upper airway angioedema compared to omapatrilat [10]. Similarly, ARNIs (e.g., sacubitril/valsartan and thiorphan/irbesartan) produced a larger (dose-dependent) blood-pressure reduction in hypertensive patients/animals compared to ARB alone, without significant differences in adverse events [11,12,13]. Furthermore, sacubitril/valsartan (LCZ696) produced a notable (24-h) blood-pressure reduction with no incidence of angioedema in randomized placebo-controlled trials (RCTs) of 1215 patients with mild-moderate hypertension [14] and in 362 hypertensive Asians [15]. Altogether, these studies highlighted the high safety profile of ARNI. 

### 3.2. Heart Failure

In the PARADIGM-HF trial enrolling HF patients with reduced ejection fraction (HFrEF), sacubitril/valsartan lowered mortality and improved clinical outcomes (e.g., risk of hospitalization, symptoms and physical limitations) compared to enalapril, although an increased incidence of hypotension was also documented. Interestingly, mild angioedema was documented in some patients receiving sacubitril/valsartan, although the number was not statistically different from the incidence in enalapril group [16]. These beneficial effects were observed across LV ejection fraction (LVEF) spectrums [17], occurred within 6 months of therapy initiation [18] and persisted even after 18 months [19]. By contrast, initial improvements in daytime physical activity and 6-min walking test with sacubitril/valsartan in HFrEF patients were no longer significant after 12 weeks in the OUTSTEP-HF study [20]. Sacubitril/valsartan also significantly decreased plasma NT-proBNP and reverted clinical features of cardiac remodeling (i.e., improved LVEF and cardiac volume) in HFrEF patients with and without type-2 diabetes mellitus in the PROVE-HF registry [21]. However, sacubitril/valsartan was not significantly different from valsartan alone in reverting structural remodeling in patients with asymptomatic post-myocardial infarction (MI) LV systolic dysfunction [22], although the efficacy of ARNI in this subset of patients is still being investigated in the PARADISE-MI trial [23]. Sacubitril/valsartan significantly lowered cardiovascular mortality, improved clinical outcomes (e.g., rehospitalization) and markedly reduced plasma NT-proBNP concentration, without major adverse events in acute decompensated HF, irrespective of the HF history or prior treatment with RAAS blockers [24,25]. Conversely, the effects of ARNI in HF patients with preserved ejection fraction (HFpEF) appear less pronounced. In PARAGON-HF, sacubitril/valsartan did not significantly lower cardiovascular mortality and hospitalization risk in the overall population [26], although a post-hoc stratification by sex revealed that sacubitril/valsartan may significantly improve clinical endpoints in women with HFpEF [27]. Additionally, sacubitril/valsartan also better preserved renal function than RAAS blockers alone, particularly in elderly and HFpEF patients [28]. 

### 3.3. Other Cardiovascular Diseases

In rats with right ventricular (RV) pressure overload and pulmonary hypertension, sacubitril/valsartan improved RV biomechanical properties and prevented RV structural remodeling [29,30]. Meanwhile, in portal hypertensive rats, sacubitril/valsartan lowered portal pressure, downregulated hepatic endothelin-1 protein expression, and reduced mean arterial pressure and systemic vascular resistance [31]. Interestingly, sacubitril/valsartan was not superior to valsartan alone in reducing blood pressure, aortic atherosclerosis and abdominal aortic aneurysm in angiotensin II-infused low-density-lipoprotein receptor-deficient mice [32], perhaps because exogenous angiotensin II was the major pathophysiological trigger in this model. Sacubitril/valsartan also significantly lowered plasma triglycerides in HFpEF patients, especially in those with higher baseline triglycerides, while modestly increasing low- and high-density lipoprotein cholesterols [33]. Moreover, one-time sacubitril/valsartan elevated the post-prandial glucagon in healthy individuals, while over 8 weeks, the drug increased fasting glucagon in obese hypertensive individuals, without changes in fasting glucose or amino acids [34], suggesting a potential role of ARNI in modulating dyslipidemia and metabolic diseases (Figure 3). 

## 4. Sacubitril/Valsartan Modulates the Propensity for Cardiac Arrhythmias in Clinical and Experimental Studies

There is a bidirectional interaction between HF and cardiac arrhythmias. For example, atrial fibrillation (AF) and HF often co-exist and influence each other’s progression [35]. On the one hand, the AF-induced loss of “atrial kick” could impair the ventricular preload, requiring extra efforts from the ventricles to maintain cardiac output, promoting ventricular remodeling and (worsening) HF. Similarly, rapid irregular ventricular activation during AF can promote a tachycardiomyopathy that may worsen ventricular remodeling. On the other hand, the HF-associated hemodynamic changes can promote excessive atrial stretch, initiating atrial electrical, structural and calcium-handling remodeling [36]. In the ventricles, HF-related structural remodeling could furthermore create an arrhythmogenic substrate, promoting reentrant ventricular arrhythmias. Therefore, HF medications can modulate arrhythmia susceptibility via direct and indirect (cardiovascular disease-related) mechanisms [37]. Indeed, both clinical and experimental studies have identified potential pro- and antiarrhythmic effects of ARNI, which are discussed below and summarized in Table 1. 

### 4.1. The Effects of ARNI on Atrial Arrhythmias

In HFrEF patients with cardiac implantable electronic devices (CIEDs), two studies have suggested a lower recurrence of atrial arrhythmias, reduced atrial arrhythmia burden and fewer ventricular extrasystoles in patients with non-permanent AF treated with sacubitril/valsartan [38,39]. However, this reduction in AF burden was not observed in another study [40]. In PARADIGM-HF, the incidence of de novo AF in HFrEF patients receiving sacubitril/valsartan (3%) did not differ from those who received enalapril [16], whereas in HFpEF patients (PARAGON-HF), sacubitril/valsartan modestly increased the incidence of new-onset AF in women, although a non-significant trend in the opposite direction was observed in men [27]. However, these latter studies were not designed to specifically address the effect of ARNI on cardiac arrhythmias and the actual incidence and burden of AF may have been underestimated since not all patients underwent continuous rhythm monitoring. 

### 4.2. The Effects of ARNI on Ventricular Arrhythmias

The incidence of sudden cardiac death, ventricular arrhythmias and appropriate implantable cardioverter defibrillator (ICD) therapy was lower in HFrEF patients with sacubitril/valsartan compared to RAAS blockers [41,42,43,44]. In agreement, a 12-month “real-world” observational study of 167 patients with dilated cardiomyopathy due to ischemic and non-ischemic origins documented a markedly diminished incidence of ICD-detected atrial and ventricular arrhythmias following sacubitril/valsartan, together with a reduction in appropriate ICD shocks [45]. This was accompanied by a significant increase of LVEF and reduction of other echocardiographic parameters (e.g., LV end-systolic and -diastolic volumes, left- and right-atrial volume index, E/A ratio and systemic pulmonary arterial pressure) following sacubitril/valsartan [45], suggesting that antiarrhythmic effects may partly reflect an ARNI-induced reverse cardiac remodeling.

By contrast, ventricular tachyarrhythmia risk did not improve after 12-months of ARNI therapy in an observational study of 59 HFrEF patients [46]. Nonetheless, the interpretation of such prospective observational data may be confounded by the natural progression of arrhythmia substrates over time, increasing the arrhythmogenic risk. Furthermore, the fact that sacubitril/valsartan did not significantly improve LVEF and HF class after 12 months [46] could be indicative for limited ARNI-induced reverse remodeling, possibly due to the ischemic origin of the HFrEF in the majority (60%) of patients [22]. Additionally, a relatively high incidence of ventricular arrhythmic storm has also been documented shortly after sacubitril/valsartan initiation in 19/218 (8.7%) HFrEF males previously on RAAS blockers [47]. Most patients had a history of ischemic heart disease (11/19), ventricular arrhythmias (12/19) and ICD (17/19) [47]. Nevertheless, whether ARNI was the cause of the arrhythmic storm remains unknown. Taken together, most studies suggest that ARNI therapy might exert antiarrhythmic effects (Table 1).

### 4.3. Preclinical Studies Investigating the Electrophysiological Consequences of ARNI

Sacubitril/valsartan partially rescued the reduction of L-type calcium current (I_Ca,L_), atrial ERP-shortening, atrial enlargement, myocardial fibrosis and enhanced AF inducibility produced by rapid-atrial pacing (RAP) in a rabbit model of AF [48]. In agreement, sacubitril/valsartan restored the ANP levels in ventricular tachypacing-induced HF rabbits post left atrial-appendage closure and normalized the prolonged atrial and ventricular effective refractory periods (ERP). Moreover, ARNI also modulated the expression levels of calcium-handling proteins and lowered the inducibility of AF and ventricular fibrillation in those animals [49]. Sacubitril/valsartan also reduced the inducibility of ventricular arrhythmias in SHRs in association with improved cardiac function and reduced electrical and structural remodeling, including normalization of action potential duration (APD) [50]. In the setting of chronic MI and HF, sacubitril/valsartan-treated rabbits also consistently displayed shorter APD, faster conduction velocity and reduced ventricular arrhythmia inducibility [51,52]. Similarly, in rats with MI-induced HF, sacubitril/valsartan attenuated the HF-induced ventricular ERP prolongation through transcriptional regulation of cardiac potassium channels [53], while in rats with ischemic cardiomyopathy, ARNI lowered ventricular arrhythmia inducibility, attenuated sympathetic neural remodeling, reversed myocardial fibrosis and increased connexin-43 expression [54]. 

In addition to reversing cardiac remodeling during long-term treatment, ARNI may also have direct antiarrhythmic effects. For example, in human ventricular cardiomyocytes from patients with end-stage HF, the combination of sacubitrilat (LBQ657) and valsartan produced a significant reduction in calcium-spark frequencies, amplitude, duration and sarcoplasmic reticulum (SR) calcium leak compared to untreated controls [55]. Additionally, sacubitrilat/valsartan partially rescued the rapid pacing-induced intracellular calcium overload in HL-1 cells [48].

## 5. Potential Mechanisms through Which ARNI Modulate Cardiac Arrhythmias

The initiation and maintenance of cardiac arrhythmias are primarily mediated by ectopic activity and reentry [36]. Briefly, ectopic activity is an uncoordinated impulse generation outside of the physiological activation sequence, while reentry is initiated when an activation wavefront propagates around anatomical or functional obstacles and re-excites the site of origin [36]. Ectopic activity is often triggered by early or delayed afterdepolarizations (EAD or DAD, respectively). Calcium-handling abnormalities, e.g., due to ryanodine receptor (RyR2) overactivation or SR calcium overload may predispose to DADs, whereas EADs are promoted by excessive APD prolongation and I_Ca,L_ reactivation. At the tissue level, APD/ERP-shortening and structural remodeling, leading to slow, heterogeneous conduction, provide a substrate for reentrant arrhythmias. 

During RAAS stimulation, angiotensin II-dependent receptor activation (i.e., AT1R) in cardiomyocytes activates protein kinase C (PKC)-dependent signaling cascades (Figure 4) [36]. During this process, phospholipase C hydrolyses phosphatidylinositol 4,5-bisphosphate into inositol trisphosphate (IP_3_) and diacylglycerol, which further activates PKC and promotes the activation of nuclear IP_3_ receptors, Ca^2+^/calmodulin-dependent protein kinase-II (CaMKII), and phosphorylation of troponin I and T, altering myofilament calcium sensitivity [36]. CaMKII is itself a major signaling molecule that affects numerous targets within a cardiomyocyte. Both PKC and CaMKII also control transcriptional regulation of ion channels and other regulatory proteins within cardiomyocytes, e.g., via nuclear calcium, histone deacetylase complexes and nuclear factor of activated T-cells (NFAT) signaling. The functional consequences of PKC activation are vast. In healthy myocardium, PKC activation increases contractility, calcium cycling and cardiac function. However, in HF, PKC promotes fibroblast proliferation and migration, as well as collagen deposition and secretion of inflammatory cytokines, facilitating structural remodeling and hypertrophy [56]. Additionally, the overstimulation of RAAS and the prolonged activation of PKC-dependent signaling cascades could facilitate cardiac arrhythmia via CaMKII-mediated phosphorylation of calcium-handling proteins, notably RyR2 and phospholamban (PLN). The PLN phosphorylation releases its inhibition of the sarco-endoplasmic reticulum calcium ATPase (SERCA), increasing calcium reuptake to the SR and promoting SR calcium overload, while the phosphorylation of RyR2 increases the open probability of the channel and promotes proarrhythmic spontaneous calcium releases and DADs [36]. In agreement with these numerous potential proarrhythmic consequences of RAAS activation, ACEi and ARB exhibit both direct and indirect (reverse remodeling-induced) antiarrhythmic effects [57,58].

NP-receptor stimulation similarly has numerous downstream effects (Figure 4). NPR-A promotes cyclic guanosine monophosphate (cGMP) production by guanylyl cyclase. Subsequently, cGMP activates protein kinase G (PKG), which has several major targets, including increased SR calcium uptake via PLN phosphorylation, inhibition of I_Ca,L_-mediated calcium influx, inhibition of PKC and hypertrophic signaling, and the reduction of myocardial stiffness and myofilament sensitivity via phosphorylation of titin and troponin I [59]. The inhibition of neprilysin promotes the activation of NP receptors, including NPR-A, subsequently reducing contractility, hypertrophic remodeling and intracellular calcium levels. Moreover, because PKG also inhibits PKC, the downstream effects of PKC are also inhibited, including the proarrhythmic calcium-handling abnormalities [55]. Additionally, NP-dependent activation of NPR-C blocks adenylyl cyclase and its downstream effects, including PKA-dependent signaling pathways. Thus, neprilysin inhibitors potentially exert strong antiarrhythmic effects. 

Since neprilysin also degrades other vasoconstriction-promoting proteins (e.g., angiotensin II and endothelin I), neprilysin inhibition may increase circulating angiotensin II levels. As such, the addition of RAAS blockers could facilitate further vasodilation, blood pressure lowering and afterload reduction, which might suppress cardiac remodeling, thereby indirectly targeting the arrhythmogenic substrate. Besides the general antiarrhythmic effects associated with RAAS and neprilysin inhibitors, ARNI (sacubitril/valsartan) appear to have additional disease-specific antiarrhythmic effects. Sacubitril/valsartan blunted the RAP-induced dephosphorylation of NFAT, downregulation of Ca_V_1.2, and upregulation of collagen I/III, NT-proBNP, ST2 and calcineurin in rabbits [48]. The consequent APD/ERP-prolongation may reduce the likelihood of reentry, decreasing the vulnerability to AF. By contrast, in HF, sacubitril/valsartan appears to reduce APD/ERP by increasing I_Kr_ and I_Ks_ through transcriptional regulation of KCNH2, KCNE1 and KCNE2 [53], potentially reducing the likelihood of proarrhythmic EADs. Moreover, it may suppress cellular determinants of ectopic activity via the downregulation of RyR2, NCX1 and CaMKII phosphorylation [51], although further studies are required to confirm the exact mechanisms and therapeutic efficacy. Finally, sacubitril/valsartan improves conduction velocity [51], which may also contribute to the reduced VF inducibility in HF. Although the abovementioned molecular mechanisms of ARNI are plausible potential contributors to the observed in vivo antiarrhythmic effects, there is at present no conclusive mechanism underlying ARNI-mediated arrhythmia suppression in patients.

Intriguingly, some studies have reported incidences of arrhythmias after sacubitril/valsartan treatment (Table 1). Although the ischemic origin of the HFrEF has been implicated in one study [47], experimental studies in ischemia-induced HF showed a reduced arrhythmia risk [51,53]. The absence of functional improvement and reverse remodeling in these patients could enable disease progression, thereby increasing the vulnerable substrate for cardiac arrhythmias. More research is needed to unravel the exact pathophysiology and mechanisms of ARNI-related proarrhythmic events. 

## 6. Conclusions

RAAS and NPs play an important role in regulating renal and cardiovascular functions in health and disease. At present, the sacubitril/valsartan combination appears to be a potent pharmacological agent to treat several cardiovascular diseases in which RAAS is overactive, including HF. Accumulating data suggest that ARNI may have antiarrhythmic effects, either by limiting cardiovascular-disease-related proarrhythmic remodeling or through direct antiarrhythmic effects on cardiomyocytes. ARNI modulate cardiac electrophysiology at various scales and affect several determinants of both atrial and ventricular tachyarrhythmias. Nevertheless, additional experiments are needed to unravel the complex mechanisms of ARNI in modifying cardiovascular pathophysiology, and to assess potential proarrhythmic effects that have also been reported. Such data are needed to gain more insights on the effects of ARNI in cardiovascular system, which will further improve the clinical application of this drug class. 

## Figures and Tables

**Figure 1 ijms-22-08994-f001:**
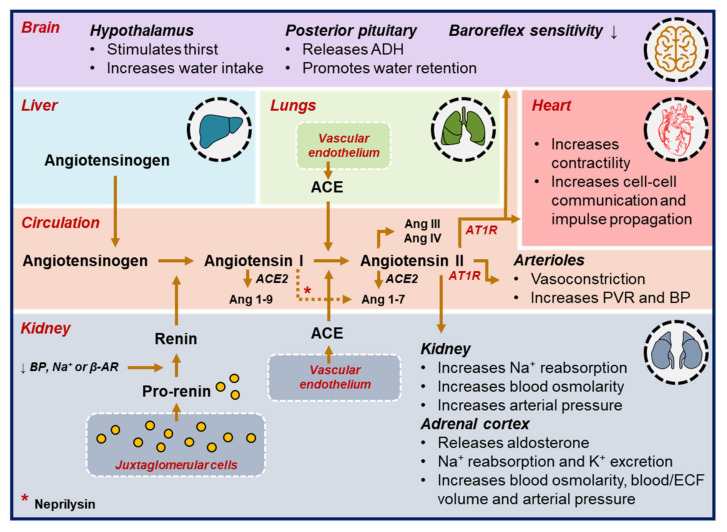
The renin-angiotensin-aldosterone activation cascade. Several stimuli including a decrease in blood pressure (BP), reduced sodium (Na^+^) levels, or activation of the autonomic nervous system promote the conversion of pro-renin into renin, which then cleaves angiotensinogen into angiotensin I. Subsequently, angiotensin I is converted into angiotensin II and binds to angiotensin receptors in brain, heart, kidney, adrenal cortex and arterioles to retain water, reabsorb sodium and initiate vasoconstriction. (ACE = angiotensin-converting enzyme; ADH = antidiuretic hormone; Ang = angiotensin; AT1R = angiotensin receptor type 1; ECF = extracellular fluid; K^+^ = potassium; Na^+^ = sodium; PVR = peripheral vascular resistance; β-AR = β-adrenergic receptor activation).

**Figure 2 ijms-22-08994-f002:**
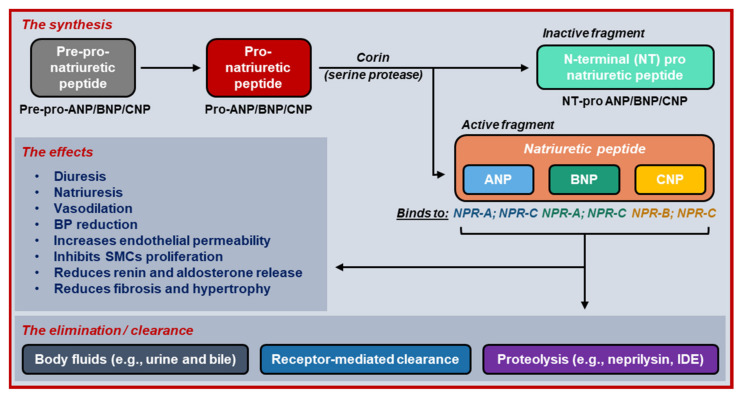
The life cycle of natriuretic peptides. Natriuretic peptides are synthesized and secreted by atrial cardiomyocytes (ANP), ventricular cardiomyocytes (BNP) and vascular endothelium (CNP) in their inactive pre-pro-form. In the presence of stimuli, the pre-pro-natriuretic peptides are converted into pro-natriuretic peptides and then cleaved by corin into both active and inactive fragments. After exerting its effects, natriuretic peptides are eliminated via three mechanisms, including degradation by neprilysin, a membrane-bound proteolytic enzyme. (ANP = atrial natriuretic peptide; BNP = brain natriuretic peptide; BP = blood pressure; CNP = C-type natriuretic peptide; IDE = insulin-degrading enzyme; NPR = natriuretic peptide receptor; SMC = smooth muscle cell).

**Figure 3 ijms-22-08994-f003:**
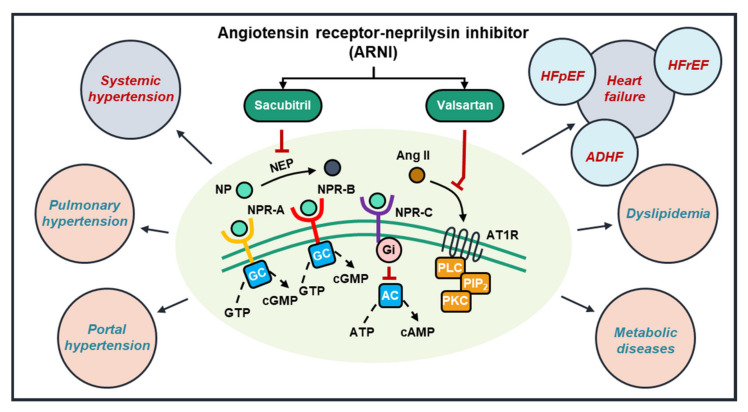
The promising roles of ARNI in cardiovascular diseases. ARNI (e.g., sacubitril/valsartan) inhibits neprilysin (NEP) and the binding of angiotensin II to its receptor (AT1R), preventing the activation of intracellular signaling cascades. In experimental and clinical studies, ARNI consistently lower blood pressure in hypertensive subjects and improve mortality and clinical outcomes in HF, particularly HFrEF (blue circles). Additionally, several studies have documented potentially beneficial roles of ARNI in other cardiovascular and metabolic diseases, warranting further investigations (pink circles). (AC = adenylyl cyclase; ADHF = acute decompensated HF; Ang II = angiotensin II; ATP = adenosine triphosphate; AT1R = angiotensin receptor type 1; cAMP = cyclic adenosine monophosphate; cGMP = cyclic guanosine monophosphate; GC = guanylyl cyclase; Gi = inhibitory G-protein; GTP = guanosine triphosphate; NEP = neprilysin; NP = natriuretic peptide; NPR = natriuretic peptide receptor; PIP_2_ = phosphatidylinositol bisphosphate; PLC = phospholipase C; PKC = protein kinase C).

**Figure 4 ijms-22-08994-f004:**
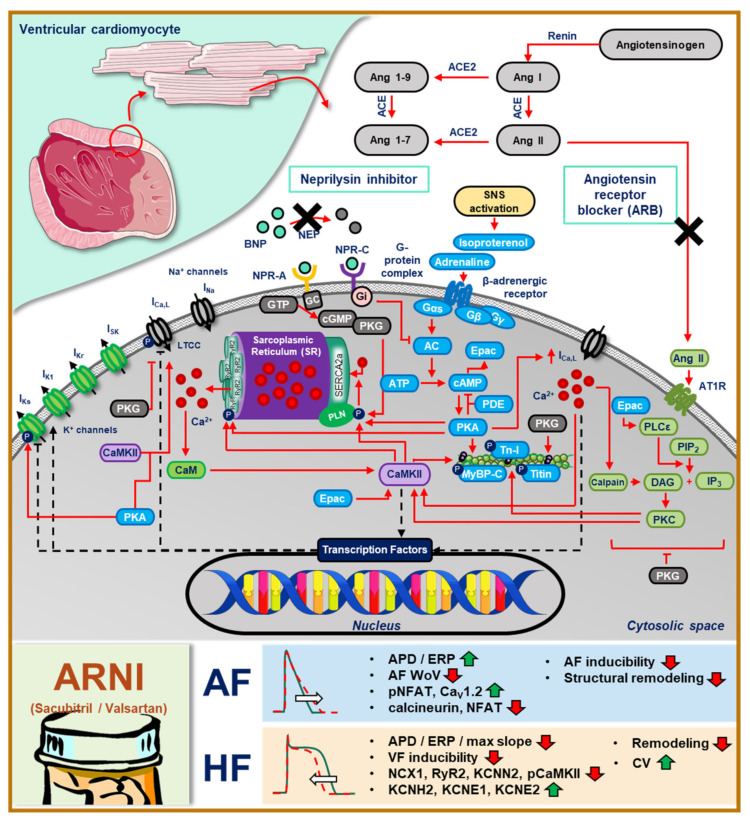
The overview of calcium-dependent signaling cascades in ventricular cardiomyocytes and the beneficial effects of ARNI in modulating cardiac electrophysiology. (AC = adenylyl cyclase; ACE = angiotensin converting enzyme; Ang II = angiotensin II; APD = action potential duration; ATP = adenosine triphosphate; CaM = calmodulin; CaMKII = calmodulin-dependent protein kinase II; cAMP = cyclic adenosine monophosphate; CV = conduction velocity; DAG = diacyl glycerol; ERP = effective refractory period; GC = guanylyl cyclase; IL = interleukin; IP_3_ = inositol trisphosphate; MyBP-C = myosin binding protein C; NEP = neprilysin; NFAT = nuclear factor of activated T-cells; NPR = natriuretic peptide receptor; PDE = phosphodiesterase; PIP_2_ = phosphatidylinositol bisphosphate; PKA = protein kinase A; PKC = protein kinase C; PKG = protein kinase G; PLC = phospholipase C; Tn-I = troponin I; WoV = window of vulnerability).

**Table 1 ijms-22-08994-t001:** Clinical evidence on the proarrhythmic and antiarrhythmic effects of ARNI.

Patient Characteristics	Number of Patients	Arrhythmic Outcome(s)	Effect	REF
Atrial Arrhythmias				
SAVETHERHYTHM (HFrEF patients with CRT-D or ICD)	Preliminary: 60 patients with sacubitril/valsartan followed over 12 months	Mean number of sustained atrial tachycardia (AT) or AF episodes per month, total AT/AF burden, mean number of premature ventricular contraction (PVC) per hour and percentage of biventricular pacing per day in patients with CRT-D	Anti-arrhythmic (lower incidence and burden of atrial arrhythmias in patients with no prior AF or non-permanent AF treated with sacubitril/valsartan)	[38]
De Vecchis et al., (NYHA class II or III chronic HF; in sinus rhythm with history of non-permanent AF; no myocardial infarction, no systemic or pulmonary embolism in the last 6 months or no absolute contraindication of oral anticoagulation)	80 patients (40 with sacubitril/valsartan and 40 with RAAS blockers)	The risk of AF relapses over 12 months	Anti-arrhythmic (Lower risk of AF recurrences with sacubitril/valsartan than with RAAS blockers)	[39]
PARADIGM-HF (NYHA class II-IV HFrEF, LVEF ≤ 40%)	8442 (4187 with sacubitril/valsartan and 4212 with enalapril)	Incidence of new-onset AF	No Effect	[16]
PARAGON-HF (NYHA class II-IV HFpEF, LVEF ≥ 45% within last 6 months, elevated NP, evidence of structural heart disease and diuretic therapy)	4796 (2479 women and 2317 men were randomized to sacubitril/valsartan and valsartan)	Incidence of new-onset AF	Pro-arrhythmic (84/1241 or 6.8% women with sacubitril/valsartan developed new-onset AF [adjusted HR 1.43 (1.02–1.99)]; 53/1166 or 4.5% men with sacubitril/valsartan [adjusted HR 0.78 (0.54–1.12)])	[27]
**Ventricular arrhythmias**				
Martens et al., (NYHA class II–IV HFrEF, LVEF ≤ 35% pre-treated with RAAS blockers; ICD or CRT implanted ≥ 6 months before sacubitril/valsartan)	151 patients with sacubitril/valsartan followed over 12 months	Device-registered arrhythmic-events (VT/VF, appropriate therapy, non-sustained VT [>4 beats and <30 s], hourly PVC burden); AF burden	Anti-arrhythmic (lower degree of VT/VF and ICD interventions but no impact on AF burden)	[40]
Rohde et al., in PARADIGM-HF (NYHA class II–IV HFrEF, LVEF ≤ 35% in the previous 6 months with elevated BNP or NT-proBNP)	8399 patients (1243 with ICD and 7156 with no ICD)	Sudden cardiac deaths (SCD) and all-cause mortality	Anti-arrhythmic (reduced SCD risk regardless of ICD status)	[41]
Russo et al., (NYHA class II DCM patients with LVEF ≤ 40% and dual-chamber ICD on sacubitril/valsartan)	167 patients with ischemic and non-ischemic DCM followed over 12 months	The incidence of device-detected atrial and ventricular tachyarrhythmia and the change in ICD parameters.	Anti-arrhythmic (reduced atrial and ventricular arrhythmias incidence and improvement of atrial ICD parameters, e.g., *p*-wave amplitude, atrial pacing threshold and atrial lead impedance)	[45]
De Diego et al., (NYHA class II–IV HFrEF, LVEF ≤ 40% with ICD and remote monitoring)	120 patients (9 months on RAAS blockers, beta blockers, MRA before RAAS blockers were switched to sacubitril/valsartan and followed for 9 months)	Appropriate shocks, non-sustained VT, PVC burden, and biventricular pacing percentage	Anti-arrhythmic (decreased ventricular arrhythmias and appropriate ICD shocks compared to RAAS blockers)	[42]
Valentim Gonçalves et al., (NYHA class II-IV HFrEF, LVEF ≤ 40% under optimized standard care—at least 6 months on RAAS blockers, beta blockers, MRA; ICD and/or CRT)	35 patients with sacubitril/valsartan followed for 6 months	ECG parameters and mechanical dispersion index, assessed by LV global longitudinal strain (GLS)	Anti-arrhythmic (reduction in QTc interval, QRS duration and mechanical dispersion index)	[43]
Gul et al., (NYHA class II–IV HFrEF, LVEF ≤ 35% with ICD and at least 6 months on RAAS blocker/beta-blockers/ivabradine/MRA, ventricular pacing percentage ≤ 40% or non-pacemaker dependent; no CRT, no R/LVH, no R/LBBB, no AF, no class I or III AADs, no end-stage renal/hepatic failure)	76 patients with sacubitril/valsartan followed for 9 months	Clinical, echocardiographic, ECG (Tp-e interval, Tp-e/QT ratio and Tp-e/QTc ratio) and device data	Anti-arrhythmic (improvement of QT interval and Tp-e related indices on surface ECG, also reduction of appropriate ICD shocks)	[44]
El-battrawy et al., (NYHA class II–IV HFrEF, LVEF ≤ 40% with ICD, CRT, pacemaker or loop recorder)	59 patients with sacubitril/valsartan, followed over 12 months	Incidence of VT and/or VF	No Effect	[46]
Vicent et al., (HFrEF without additional specific criteria)	218 patients with sacubitril/valsartan	N/A(this is a case-series)	Pro-arrhythmic (19/218 or 8.7% developed sustained ventricular arrhythmias. 19/19 received RAAS blockers before ARNI, 18/19 with beta-blockers, 17/19 were in ICD, 12/19 had a history of ventricular arrhythmias and 5/19 had an episode of VT/VF within 6 months prior to ARNI)	[47]

## Data Availability

Not applicable (This manuscript does not include any new data).

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
