# Peer review of "Angiotensin Receptor-Neprilysin Inhibitor (ARNI) and Cardiac Arrhythmias"

_ijms, 2021, doi:10.3390/ijms22168994_

Round 1

Reviewer 1 Report

This review by Sutanto and colleagues focuses on the antiarrhythmic properties of angiotensin receptor-neprilysin inhibitors (ARNI). Their detailed overview of clinical trials suggests that the combination of type I angiotensin receptor blocker and neprilysin inhibitor confers benefits to patients with heart failure and patients with systemic, pulmonary or portal hypertension. Specifically, the risk reduction for atrial and ventricular arrhythmias may contribute to the overall clinical benefit. Although a number of more recent experimental animal studies have investigated direct electrophysiological effects of ARNIs and their underlying molecular mechanisms at the whole-animal and single cell level, our understanding of the underlying signaling pathways remains incomplete. The authors list possible signaling pathways by which activation of NP receptors may reduce arrhythmogeneity (lines 294-379). Although  potentially interesting, it is unclear what is actually known and what is merely speculation regarding the molecular events responsible for ARNI-mediated arrhythmia suppression. The authors may want to distinguish potential vs. proven mechanisms in the section 5 of this otherwise excellent, carefully written manuscript.

Author Response

This review by Sutanto and colleagues focuses on the antiarrhythmic properties of angiotensin receptor-neprilysin inhibitors (ARNI). Their detailed overview of clinical trials suggests that the combination of type I angiotensin receptor blocker and neprilysin inhibitor confers benefits to patients with heart failure and patients with systemic, pulmonary or portal hypertension. Specifically, the risk reduction for atrial and ventricular arrhythmias may contribute to the overall clinical benefit. Although a number of more recent experimental animal studies have investigated direct electrophysiological effects of ARNIs and their underlying molecular mechanisms at the whole-animal and single cell level, our understanding of the underlying signaling pathways remains incomplete.

We would like to thank the reviewer for his/her time and willingness to critically review our manuscript. We fully agree with the reviewer that the molecular mechanisms underlying the potential pro- or antiarrhythmic effects of ARNI remain incompletely elucidated and hope that our review could provide a useful state-of-the-art summary of both the clinical evidence and potential molecular targets of ARNI in cardiomyocytes.

The authors list possible signaling pathways by which activation of NP receptors may reduce arrhythmogeneity (lines 294-379). Although potentially interesting, it is unclear what is actually known and what is merely speculation regarding the molecular events responsible for ARNI-mediated arrhythmia suppression. The authors may want to distinguish potential vs. proven mechanisms in the section 5 of this otherwise excellent, carefully written manuscript.

Thank you for this excellent suggestion. In Section 5, particularly in lines 354-364, we have summarized the known data from available experimental studies about the direct and indirect electrical remodeling by ARNI, and linked them with the known / reported data on the molecular mechanisms of RAAS and NPR stimulation on cardiac arrhythmias.

Sacubitril/valsartan blunted the RAP-induced dephosphorylation of NFAT, downregulation of CaV1.2, and upregulation of collagen I/III, NT-proBNP, ST2 and calcineurin in rabbits [48]. The consequent APD/ERP-prolongation reduces the likelihood of reentry, decreasing the vulnerability to AF. By contrast, in HF, sacubitril/valsartan appears to reduce APD/ERP by increasing IKr and IKs through transcriptional regulation of KCNH2, KCNE1 and KCNE2 [53], potentially reducing the likelihood of proarrhythmic EADs. Moreover, it may suppress cellular determinants of ectopic activity via the downregulation of RyR2, NCX1 and CaMKII phosphorylation [51], although further studies are required to confirm the exact mechanisms and therapeutic efficacy. Finally, sacubitril/valsartan improves conduction velocity [51], which may also contribute to the reduced VF inducibility in HF

Although the abovementioned molecular mechanisms of ARNI are plausible potential contributors to the antiarrhythmic effects observed in vivo in animal / clinical studies, at present, there is no proven and validated mechanism responsible for ARNI-mediated arrhythmia suppression in patients. Indeed, the major aim of our review was to recapitulate the available (currently rather fragmented) data and propose putative mechanisms that are worth looking into in future experiments. However, given the numerous direct and indirect effects of ARNI on the heart it may be very challenging to dissect which molecular target is primarily responsible for any antiarrhythmic effects. Based on the reviewer’s valuable suggestion, we have further emphasized this lack of direct evidence in the revised version of the manuscript:

Page 1, Section 1: “Here, we review the putative roles of ARNI in cardiovascular disease management, focusing on their potential effects in cardiac arrhythmogenesis, and discuss the potential molecular mechanisms of arrhythmia modulation by ARNI.

Page 11, Section 5: “Although the abovementioned molecular mechanisms of ARNI are plausible potential contributors to the observed in vivo antiarrhythmic effects, there is at present no conclusive mechanism underlying ARNI-mediated arrhythmia suppression in patients.

Reviewer 2 Report

Using an organized and easy-reading text, the authors discuss the effectiveness of the combination of angiotensin receptor blocker and neprilysin inhibitor (ARNI) and its association with atrial and ventricular arrhythmias, providing novel insights about the therapeutic role and safety profile of ARNI in the cardiovascular system.

The use of very informative and self-explanatory schematics facilitates the understanding and places the reader into the problem and provide a great tool for investigators interested in the topic.

Author Response

Using an organized and easy-reading text, the authors discuss the effectiveness of the combination of angiotensin receptor blocker and neprilysin inhibitor (ARNI) and its association with atrial and ventricular arrhythmias, providing novel insights about the therapeutic role and safety profile of ARNI in the cardiovascular system.

The use of very informative and self-explanatory schematics facilitates the understanding and places the reader into the problem and provide a great tool for investigators interested in the topic.

We would like to thank the reviewer for her/his effort to review our manuscript, and for the overall positive evaluation.